# Effects of Male and Female Strains of *Salix linearistipularis* on Physicochemical Properties and Microbial Community Structure in Saline–Alkali Soil

**DOI:** 10.3390/microorganisms11102455

**Published:** 2023-09-29

**Authors:** Haojun Cui, Yan Li, Wenyi Wang, Lili Chen, Zhouqing Han, Shurong Ma, Weidong Wang

**Affiliations:** Key Laboratory of Saline-Alkali Vegetation Ecology Restoration, Ministry of Education, College of Life Science, Northeast Forestry University, Harbin 150040, China

**Keywords:** woody plant gender, saline–alkali land, rhizospheric microorganism, soil physicochemical properties, *Salix linearistipularis*

## Abstract

The woody plant gender difference may lead to alteration in rhizosphere microbial communities and soil physicochemical properties. In this study, we investigated the differences in rhizosphere soil properties and microbial community structures of *S. linearistipularis*. Rhizosphere microorganisms were analyzed by high-throughput sequencing technology. The results showed that there were significant differences in rhizosphere soil nutrition between male and female *S. linearistipularis* plants in saline–alkali soil. The female *S. linearistipularis* plants significantly reduce soil pH values and significantly increase the soil water content (SWC), available total nitrogen (TN), soil organic matter (SOM), and soil urease activity (S-UE) compared to the male plant. The ACE, Chao, and Shannon index of the female plant was significantly higher than that of the male strain. At the level of Bacteriophyta, the relative abundance of Actinobacteriota in male and female *S. linearistipularis* was the highest, with 34.26% and 31.03%, respectively. Among the named bacterial genera, the relative abundance of *Defluviicoccus* of male and female plants was the highest, with 2.67% and 5.27%, respectively. At the level of Eumycophyta, the relative abundance of Ascomycetes in male and female plants was the highest, with 54.93% and 52.10%, respectively. Among the named fungi genera, the relative abundance of male and female plants of *Mortierella* was the highest, with 6.18% and 9.31%, respectively. In addition, soil pH, SOM, SWC, and S-UE activities were the main driving factors of soil microbial community structures. In the process of restoring saline–alkali land in the Songnen Plain, we may prioritise the planting of female *S. linearistipularis*, which also provides a theoretical basis for the microorganisms restoration of saline–alkali land in the Songnen plain.

## 1. Introduction

Soil salinization is a serious problem and restricts agricultural production and sustainable land use on the Songnen plain [1]. The Songnen Plain is not only an important grain production region, but also holds the main concentration of saline–alkali land worldwide [2]. Recently, due to natural and man-made factors, soil salinization in the Songnen Plain put great pressure on the limited soil resources and had a negative impact on the ecosystem [3,4]. Therefore, restoring the saline–alkali land to arable land is of great significance in ecological restoration and economic development.

The plants play an important role in biodiversity conservation and terrestrial ecosystem function [5,6]. For example, *S. linearistipularis* is a common dioecious plant with strong salt–alkali tolerance, cold tolerance, and drought tolerance [7]; these factors are important for improving saline–alkali soil and ecological restoration. Male and female plants have significant differences in morphology and physiology [8,9].

Rhizosphere microorganisms play a key role in plant growth and environmental adaptation, and their communities are influenced by root–soil interactions [10,11]. The exchange of matter and energy between soil, rhizosphere microorganisms, and plants form a close relationship, which make the abundances and species of rhizosphere microorganisms differ from that of non-rhizosphere soils [12]. In addition to genetic factors, plant salt and alkali tolerance is also determined by microorganisms in the rhizospheric ecosystem [13,14].

Salinization is a major threat to soil microorganisms and plants and has negative effects on organisms due to water shortage, as well as decreasing osmotic potential and ionic toxicity [15]. Rhizosphere microorganisms can improve plant salt tolerance by regulating the ion homeostasis of plant roots, producing plant hormones, promoting the accumulation of osmotic substances in roots, and enhancing plant antioxidant activity and nutrient absorption [16]. *S. linearistipularis* is a host plant for mycorrhiza formation, and the mycorrhiza formed by its roots can reduce the pH value of soil around the roots and effectively improve the saline–alkali soil environment [17]. However, the research community has limited knowledge on the rhizosphere microbial communities and microenvironments of the dioecious plant *S. linearistipularis* under salinization conditions. In this study, we aimed to investigate the effects of soil physicochemical properties and sex differences of rhizosphere soil microbial communities on saline–alkali soil. In addition, we also compared the difference of male and female *S. linearistipularis* plants on saline–alkali soil improvement. Therefore, we hypothesized that the soil physical and chemical properties of *S. linearistipularis* at the same sampling site may be driving factors for the microbial community in the rhizosphere. Through the comparison of rhizosphere microbial communities of male and female *S. linearistipularis*, the theoretical basis was provided for improving saline–alkali soil more effectively through microbial measures under saline–alkali stress conditions. Overall, *S. linearistipularis* can improve saline–alkali soil; the female plant has a better ability to improve saline–alkali soil than the male plant [18].

For a long time, dioecious plants have been the focus of research in the field of botany because of the differences in various plant indexes caused by different genders. The studies of dioecious plants mainly focus on morphological and physiological differences, molecular biological differences, and stress resistance differences, while the rhizosphere microbial community differences are less studied. Zhu [19] found that gender differences affected the nutrient utilization patterns and bacterial community structure of rhizosphere soil of male and female *Populus deltoides*. The α-diversity index of rhizosphere bacterial community of male was significantly higher than that of female, and the symbiotic network of rhizosphere soil bacterial community of male *Populus deltoides* was more diverse and complex, and male *Populus deltoides* were more able to adapt to environmental changes than female *Populus deltoides*. Wu [20] found that sex differences in *Populus cathayana* would lead to differences in rhizosphere microbial communities of male and female plants, and different salt–alkali stress also had sex differences in changing rhizosphere microenvironment and microbial communities of *Populus cathayana*. At present, the reproductive mode [7], morphological structure [21], genetic transformation [22], and stress response gene function [23] of *S. linearistipularis* have been studied. However, there are no studies on the comparison of rhizosphere microbial differences between male and female *S. linearistipularis* [24]. By studying dioecious rhizosphere microorganisms of *S. linearistipularis*, we can provide typical data for the gender differences in microbial composition and species of male and female *S. linearistipularis*. At the same time, it provides reference for understanding the effects of the microbial community on the adaptation and ecological function of the dioecious plants of *S. linearistipularis*.

## 2. Materials and Methods

### 2.1. Sampling Site and Collection

The sampling area is located in Anda City, Heilongjiang Province (46°4′–47°1′ N, 124°53′–125°55′ E), China. This region belongs to the north temperate continental semi-arid monsoon climate, the average annual temperature is −4.2 °C, the average annual precipitation is 432.5 mm, and the frost-free period lasts for 129 days. The main soil type in the Anda region is saline–alkali soil. *S. linearistipularis* is a rare woody plant with a natural population in the saline–alkali soil in this region.

The rhizosphere soil was collected in October 2021. Twenty *S. linearistipularis* were divided into a male group and a female group, with ten willows in each group. The selected *S. linearistipularis* in each group had similar habitats, flourishing stages, and tree ages. The distance between two trees in the same group was 100–200 m, and the sex of the trees was identified and marked during the flowering season. Litter and humus were removed before soil sampling. A sampling area (1 m × 1 m) was set up in the center of each willow tree, and five soil sampling sites were set up in the center and four corners of the sampling area. Soil samples from five sites (*S. linearistipularis* root surface and surrounding 1–3 mm) were collected and then brought together. After mixing, the samples were divided into two parts: one sample was used to analyze soil physical and chemical properties, and another sample was stored at −80 °C.

### 2.2. Analysis of Physical and Chemical Properties of Soil

The pH meter (PHS-3C-25, Yue Ping, Shanghai, China) was used to measure soil pH. The soil water content (SWC) was calculated using the drying-weighing method [25]. Soil electrical conductivity (EC) was measured with the conductivity meter (DDS-11A, Yue Ping, Shanghai, China). Soil organic matter content (SOM) was determined using organic matter photometry. Soil total available phosphorus (TP) were quantified using a molybdenum-antimony anti-colorimetric approach. Soil total available nitrogen (TN) was quantified via an alkaline hydrolysis diffusion approach [26]. Soil total available potassium (TK) was quantified via an ammonium acetate extraction flame photometric method [27]. In accordance with the method presented by Lee et al., fresh soil was weighed (3 g) into a 50 mL triangle bottle to measure the soil sucrase activity (S-SC) [28]. Soil urease activity (S-UE) was determined according to Kandeler and Eder [29]. Soil alkaline phosphatase activity (S-AKP) was determined according to Pozo et al. [30]. Soil catalase activity (S-CAT) was determined according to Guwy et al. [31]. All the above tests were repeated three times, and the final results were averaged.

### 2.3. Soil DNA Extraction, PCR Amplification, and High-Throughput Sequencing

Soil microbial community genomic DNA was extracted from samples using the E.Z.N.A. soil DNA kit (Omega Bio-tek, Norcross, GA, USA) according to the manufacturer’s instructions. DNA extracts were examined on 1% agar–agar gel and DNA concentration and purity were determined by NanoDrop 2000 UV-vis spectrophotometer (Thermo Scientific, Wilmington, DE, USA). The extracted soil DNA was commissioned to Megi Biological Technology Co., Ltd. (Shanghai, China) for high-throughput sequencing. The abundance of microbial communities was estimated using PCR amplification. The target genes were bacterial 16S rRNA and fungal ITS. Bacterial 16S rRNA was amplified using primers 338(5′-ACTCCTACGGAGGCAGCAGCAG-3′) and 806R(5′-126GGACTACHVGGGTWTCTAAT-3′). Primers ITS1F (5′-CTTGTCATTTAGAGGAAGTAAGTAA-3′) and ITS2R(5′-GCTGCGTTCTTCATCGATGC-3′) were used.

The PCR mixture consisted of 4 μL of 5 × 129 TransStart FastPfu buffer, 2 μL of 2.5 mM dNTP, 0.8 μL of forward primers (5 μM), 0.8 μL of reverse primers (3 μM), 0.4 μL TransStart-FastPfu DNA polymerase, 10 ng template DNA, and ddH_2_O (up to 20 μL). PCR reaction was performed in triplicate. PCR products were extracted from 2% agarose gel and purified according to the manufacturer’s instructions using the AxyPrep DNA gel extraction kit (Axygen Biosciences, Union City, CA, USA) and Quantus fluorometer (Promega, Madison, WI, USA).

After sample separation of PE reads obtained by Illumina sequencing, and quality control and filtering of double-ended Reads were carried out first according to sequencing quality. Meanwhile, the optimal data after quality control and stitching were obtained according to the overlap between the double-ended Reads. The optimized data were then processed by sequential noise reduction method (DADA2/Deblur, etc.), and sequence and abundance information were obtained by ASVs (amplicon sequence variants). Based on the representative sequence and abundance information of ASVs, a series of statistical or visual analyses such as community diversity analysis, species difference analysis, and correlation analysis were performed.

### 2.4. Statistical Analysis

In this experiment, packages of R were used to analyze the rhizosphere microbial community structure of male and female *S. linearistipularis*. SPSS 26.0 statistical software was used to test the significance of differences in soil physical and chemical properties and rhizosphere microbial diversity index (*p* < 0.05) by single-factor analysis of variance and multiple comparisons. In addition, four α-diversity indices (Ace, Chao, Shannon and Simpson) were calculated for each soil compartment using Mothur (version 1.30) ver.

## 3. Results

### 3.1. Physicochemical Properties of Rhizosphere Soil of Male and Female S. linearistipularis

The contents of SOM, TN, SWC, and S-UE in soil with female *S. linearistipularis* were higher than in male (*p* < 0.05) (Table 1), while the contents of pH in male *S. linearistipularis* were significantly higher compared to the female *S. linearistipularis* (*p* < 0.05). The soil properties differences between male and female *S. linearistipularis*, suggesting that sex differences might affect rhizosphere nutrients. The organic matter content of the female was 7.66%, which was significantly higher than that of the male, which was 4.25%. The TN content of the female was 113.47 mg/kg, which was significantly higher than that of the male, which was was 76.93 mg/kg. In addition, the S-UE activity of the female strain (859.23 μg/d/g) was significantly higher than that of male strain (712.48 μg/d/g). SWC (26.90%) of the female plant was significantly higher than that of the male plant (22.68%). The pH (8.23) of the female plant was significantly lower than that of the male plant (8.43). This indicated that the female plant had better soil nutrients and better ability for improving saline–alkali soil than the male plant.

### 3.2. Rhizosphere Soil Microbial Communities between Male and Female Salix linearistipularis

The diversity indices of Chao, ACE, Simpson, and Shannon in the rhizosphere soil of male and female plants were compared (Figure 1). The diversity indices refer to the average. It was found that the ACE, Chao, and Shannon values of the rhizosphere soil bacteria were significantly different in the male and female *S. linearistipularis* (*p* < 0.05). The ACE index of the female strain was 1382.75, which was significantly higher than that of the male strain, at 1219.07. The Chao index of the female plant was 1380.52, which was significantly higher than that of the male plant, at 1217.42. The Shannon index of the female plant was 6.68, which was significantly higher than that of the male plant, at 6.49. However, there was no significant difference between male and female rhizosphere soil fungi (*p* > 0.05), but the diversity index of the female plant was higher than that of the male plant. This revealed that the soil microbial community under the female plants is more diverse than that under the male plants. Principal component analysis (PCA) indicated that the bacterial and fungal communities were different under the soil surrounding the male and female *S. linearistipularis* (Figure 2).

A Venn diagram was used to explore the common and unique soil microorganisms at ASV levels between different genders in *S. linearistipularis* (Figure 3). At the ASV level, there were 2560 bacterial ASVs in the male and female strains of *S. linearistipularis*, accounting for 46.2% and 48.6% of the total bacterial ASVs, respectively. The male and female strains had 2980 and 2170 unique bacterial ASVs, respectively. There were 1059 fungal ASVs in male and female strains, accounting for 36.6% and 36.1% of the total fungal ASVs, respectively. There were 1837 and 1874 unique fungal ASVs in male and female strains, respectively.

The bacterial and fungal community composition of male and female *S. linearistipularis* at phylum and genus levels showed that the rhizosphere soil bacteria belonging to 45 phyla, 141 classes, 304 orders, 482 families, and 867 genera. The relative abundance of the sample gate exceeds 1%. Actinomyces (31.03%) were the most common, followed by Proteobacteria (22.92%) and Acidobacteria (17.31%). Others, such as Blastomonas, accounted for 1.17% (Figure 4A). The relative abundance of the dominant bacterial phyla was different between the male and female strains. For example, Actinomyces decreased from 34.26% to 27.80% in females, Proteobacteria increased from 21.90% to 23.95%, and Acidobacteria increased from 15.96% to 18.64% as compared to the male plants. At the genus level (Figure 4B), most bacteria in the rhizosphere soil of male and female *S. linearistipularis* belong to no rank and are unclassified, with 21 genera having relative abundances exceeding 1%. Similarly, the abundance of genus *Defluviicoccus* was 2.72% and 5.27% in the rhizosphere of male and female strains, respectively. Compared with the male, the relative abundance of *Bacillus* was increased from 2.24% to 2.86% in the female *S. linearistipularis* rhizosphere, while the relative abundance of *Gaiella* decreased from 2.66% to 1.88%.

The male and female rhizosphere fungi belong to 15 phyla, 50 classes, 116 orders, 243 families, and 512 genera. Figure 4C shows the sample fungi doors’ relative abundances of more than 1%. Among them, with a relative abundance of at 53.51%, Ascomycetes door is highest, followed by the unclassified_k_Fungi door (18.05%), Basidiomycete fungi (17.76%), Mortierellomycota (7.78%), and Rozellomycota (1.87%), with the latter being relatively low. Relative to the male and female plant, the Ascomycetes door decreased from 54.93% to 52.10%, the unclassified_k_Fungi fell from 21.85% to 14.26%, and the Basidiomycete fungi door decreased from 20.92% to 14.60%. In fungi, the genera level (Figure 4D) included *g__unclassified_k__Fungi*, *Mortierella* genera, *Inocybe*, *Humicola* genera, and *Fusarium* genera, which is dominant in the sample fungi community.

### 3.3. Correlation between Microbial Communities and Soil Physicochemical Properties

Among the top 21 dominant bacterial genera with relative abundance greater than 1%, 10 in male soils and 9 in female soils are significantly correlated with the soil nutrients (Figure 5A). Similarly, among the top 18 dominant fungal genera with relative abundance greater than 1%, nine in male soils and twelve in female soils are significantly correlated with soil nutrients (Figure 5B).

There is significant correlation between the content of the TP, S-AKP, S-UE, S-CAT, pH, EC, and bacterial genera of the male *S. linearistipularis*. *Rubrobacter* is negatively correlated with TP and EC, while *Defluviicoccus* has significant positive correlations with TP and EC. In contrast, there is significant correlation between the SOM, TP, S-AKP, S-UE, S-CAT, pH, EC, and fungal genera of the female *S. linearistipularis*. *Rubrobacter* is negatively correlated with S-CAT. *Gaiella* is positively correlated with pH.

In the dominant genera of the fungal community, there is significant correlation between the SOM, TP, TK, TN, S-AKP, S-UE, EC, and fungal genera of the male *S. linearistipularis*. *Sporormiella*, *Hebeloma*, and *Serendipita* are significantly negatively correlated with TP, while *Geopora* are significantly positively correlated with TP. *Tomentella* is positively correlated with S-AKP, but is negatively correlated with S-UE. *Sporormiella*, *Tomentella*, *Hebeloma*, and *Serendipita* are significantly negatively correlated with EC, while *Geopora,* and *Neocosmospora* are significantly positively correlated with EC. There is significant correlation between the SOM, TP, TK, TN, S-AKP, S-UE, S-SC, SWC, and bacterial genera of the female *S. linearistipularis*. *Mortierella* are significantly correlated with S-AKP; *Inocybe* are significantly correlated with SOM, TP, and S-UE; *Inocybe* and *Serendipita* are significantly correlated with SOM, TP, and S-UE; *Geopora* are significantly correlated with TP; *Metarhizium* are significantly correlated with SOM, TK, TN, and S-SC; *Didymella* are significantly correlated with SOM, TK, and S-UE; *Hebeloma* are significantly correlated with TK; *Tomentella* are significantly correlated with TN, TP, and S-UE.

The redundancy analysis (RDA) method was used to study the correlation between soil microbial communities and environmental factors in the rhizosphere soil of *S. linearistipularis* (Figure 6). The results showed that S-UE was the main environmental factor that significantly affected the rhizosphere bacterial community in *S. linearistipularis*, while SWC, pH, SOM, and S-UE were the major environmental factors that significantly affected the rhizosphere fungal community (*p* < 0.05).

## 4. Discussion

### 4.1. Effects of Male and Female on Salix linearistipularis Soil Physicochemical Properties

Among *salicaceae* plants, male *Populus* plants are more sensitive and tolerant to saline–alkali-stressed environments and have better defense abilities than female plants [32,33,34]. However, the opposite phenomenon exists in *Salix* plants [7,18]. In addition, in some dioecies of other families, there is a phenomenon that female plants are more adaptable to salt stress than male plants. For example, under salt stress conditions, the photosynthetic rate, water use efficiency, and antioxidant enzyme activity of female plants of *Ginkgo biloba*, *Humulus ulus*, *Amaranthuscannabinus,* and other species are higher than those of male plants, and they have greater salt tolerance and better sex-specific strategies [35,36,37]. Dioecious plants have always been the focus of research in the field of botany because of the differences in various plant indexes caused by different genders. However, the determination of plant indexes of *S. linearistipularis* mainly focuses on the determination of leaf physiology [7,18,38], and there are few studies on the determination of rhizosphere soil physicochemical properties [24].

The relationships between plant sex and soil properties have been reported on in the literature [39,40,41,42]. In this study, *S. linearistipularis* has salt–alkali tolerance, but this is reflective of soil properties based on the different gender relations in the rhizosphere community. Wu [20] found that, under salt stress, there were significant differences in the EC and pH values of rhizosphere soil, as well as differences in soil nutrients and enzyme activities, resulting in soil heterogeneity. *S. linearistipularis* is a host plant for mycorrhiza formation, and the mycorrhiza formed by its roots can reduce the pH value of soil around the roots and effectively improve the saline–alkali soil environment [17]. The soil pH values of female *S. linearistipularis* were lower than those of the male plant, indicating that female *S. linearistipularis* may be better than male *S. linearistipularis* in improving salt–alkali soil and maintaining the relative stability of the soil microenvironment. A series of material exchanges and energy transfers between plants and soil change the physicochemical properties of the soil [43]. The female plants had higher soil water organic matters, nitrogen, phosphorus, and potassium contents compared to male *S. linearistipularis*. In addition, the female *S. linearistipularis* plants significantly increased the soil water content (SWC), available total nitrogen (TN), and soil organic matter (SOM). In the process of vegetation life activities, roots secrete organic matter into the soil, which is further mineralized and transformed into organic acids to reduce soil pH. The content of soil organic matter for the female plant was significantly higher than that of the male plant, and the soil pH value was significantly lower than that of the male plant. This may be because the root secretion ability for organic matter of the female plant was stronger than that of the male plant, and because the decrease in the soil pH range was greater than that of the male plant. Soil provides nutrients for plant growth, and vegetation growth produces litter and root exudates, which gradually accumulate through continuous decomposition and mineralization and input into the soil, returning nutrients to the soil and improving the soil nutrients [44]. Compared with male plants, the female plants of *S. linearistipularis* were taller, with wider crown shapes and higher above-ground biomass, such as branches and leaves. Therefore, female willow plants may secrete more root exudates to decompose more litter, resulting in higher levels of soil nutrients in female than male plants. Similarly, soil enzymes also play an important role in nutrient cycling and microbial metabolism in the ecosystem [45]. For example, in this study, soil urease (S-UE) in the female plants was significantly higher than those in the male plants (Table 1). Soil urease is one of the most active hydrolases in soil, which can catalyze the hydrolysis of urea in soil to produce ammonia and CO_2_ [46]. The urease enzyme might promote the biochemical synthesis of soil nitrogen and ensure the nitrogen supply capacity of the soil [47]; this theory is consistent with the results in this study: the content of soil-available nitrogen for female plants was significantly higher than that for male plants. The soil enzyme activity of the female plants was higher than that of the male plants, which may be related to the rich and diverse microbial community structure of the female plants. Soil enzymes are mostly secreted by soil microorganisms, which have a certain regulatory ability in response to stress. Salt and alkali in soil stimulate the growth or reproduction of some tolerant microorganisms and increase the synthesis and secretion of intracellular enzymes of microorganisms and extracellular enzymes secreted into soil. It may also be due to the fact that microorganisms require more energy to maintain survival under salt and alkali stress. These results indicated that the female plants improve the soil environment compared to the male plant under saline–alkali conditions.

### 4.2. Effects of Male and Female Salix linearistipularis on the Soil Microbial Community

Previous studies have shown that mycorrhizal fungi are related to the salt–alkali tolerance of *S. linearistipularis* through the analysis of the ectomycorrhiza fungal community [48], endophytic fungi [49] of *S. linearistipularis* root, and the isolated Trichoderma [50], which also proves that the rhizosphere microorganisms of *S. linearistipularis* are related to their salt–alkali tolerance.

The rhizosphere soil microbial community of host plants is closely distributed among the root system and is affected by plant species more significantly than by environmental factors [51]. However, in this study, the microbial community and microenvironment in the *S. linearistipularis* rhizosphere showed a sex-differential preference. Similarly, Actinomycetes and Proteobacteria were the most abundant bacterial phyla in the male and female *S. linearistipularis* specimens, but their relative abundance was also different between the sexes. Actinomycetes are exhibited under stress conditions and play an important role in the biodegradation and recycling of organic matter, the decomposition of organic matter, resistance against pathogens, and the maintenance of the spatial distribution stability of bacterial communities [52]. The relative abundance of Actinomycetes in the rhizosphere soil of male and female *S. linearistipularis* was the highest, and the root activities of *S. linearistipularis* played a more positive role in rhizosphere soil. Proteobacteria include many azotobacter bacteria with a wide distribution range and strong adaptability to saline–alkali rhizosphere environments [53]. Similar results were observed in this study which showed that the relative abundance of Proteobacteria in the rhizosphere soil of female *S. linearistipularis* was higher than that of male plants in saline–alkali soil. Acidobacteria are intolerant to salt and alkali, which may be the reason why the proportion of male and female strains in *Salix linearistipularis* is relatively low. Studies have proved that soil microorganisms can accumulate nutrients and reduce soil pH by releasing organic acids into the soil, and some *Acidobacterium* can degrade soil cellulose and produce acetic acid [54]. Hence, the presence of *Acidobacterium* in female strains was higher than that in male strains, improving the saline–alkali ability of female strains. The relative abundance of *Ascomycetes* in rhizosphere soil fungi of male and female *S. linearistipularis* was more than 50%, becoming the dominant phylum. The relative abundance of the dominant fungal phylum Ascomycetes was higher than that of Basidiomycetes. This may be due to the fact that Ascomycetes are more drought-tolerant and have a higher conversion rate than Basidiomycetes [55]; additionally, Ascomycetes are mostly saprophytic bacteria, which play an important role in degrading soil organic matter [56]. Most Ascomycetes can decompose lignin, keratin, and other difficult-to-degrade organic matter. Lignin cellulose is a kind of difficult-to-degrade organic matter which plays an important role in the nutrient cycle [57]. Therefore, both male and female strains of *S. linearistipularis* decomposed the organic matter. In Basidiomycetes, the genera of *Cyanocyta* and *Filamina* can establish symbiosis with plants to form mycorrhiza and enhance plant resistance [58]. The Basidiomycetes was more dominant in female strains as compared to the male strains. Therefore, it is speculated that the decrease in the salinity of the soil pH of female plants relative to male plants may be related to the higher proportion of Basidiomycetes. A finding which is consistent with previous studies is that the microbial community diversity was limited by soil salinity and available element sources [59]. Salt content can affect the availability of soil water or the metabolic activities of microorganisms, thus causing changes in soil microbial diversity [60]. Water affects the growth and vitality of plant roots, changes the content of root exudates, and affects the community structure and quantity of fungi in rhizosphere soil [61,62]. *Fusarium* can secrete cellulase, play a role in the decomposition of carbon, and participate in the dissolution of soil-insoluble phosphorus together with penicillium; this improves the uptake and utilization of phosphorus by plants, meaning that they are important phosphorus-solubilizing microorganisms [63]. The pH value of female plants was lower than that of male plants, and the relative abundance of Basidiomycetes and *Fusarium* in the female strains was higher than that in the male strains, which may explain the above conclusions. In addition, female plants with more litter can also reduce soil water evaporation, maintaining soil moisture and providing a better survival and reproductive environment for soil microorganisms. This also contributes to maintaining the richer soil microbial community possessed by female plants.

According to the present results, bacteria were the main community in the rhizosphere soil microorganisms of male and female *S. linearistipularis* plants, and the number of fungi was small, whereas the rhizosphere soil microbial diversity of male and female *S. linearistipularis* plants was closely related to the characteristics of rhizosphere soil [20]. A higher alpha diversity index is indicative of a richer microbial community species and a more stable community composition. Some previous studies [19,64] showed that soil microbial diversity and richness of female dioecious woody plants were higher than male plants under saline–alkali stress. The Shannon index and Simpson index of bacteria and fungi of female strains were relatively higher than male strains, indicating that female strains had the highest diversity of bacterial species. If the Chao index and ACE index of female plants were higher than that of male plants, then the microbial community richness of female plants was higher. These results indicated that female plants had higher microbial richness and diversity and a more stable community structure than male plants.

The interactions between microbial community composition and environmental variables indicate that soil can affect related microbial communities; additionally, they show that soil properties are the main driving factors of soil microbial communities, which are also closely related to plant sex differences [43,65].

Soil microbial communities regulate soil organic carbon decomposition and nutrient cycling [66]. Under saline–alkali conditions, the microbial community structure of the female plants was relatively stable, and the richness and diversity of the female plants were higher than those of the male plants. This may be the reason for the heterogeneity of soil and the difference in saline–alkali soil improvement ability between male and female *S. linearistipularis* plants.

### 4.3. Soil Physical and Chemical Properties of Salix linearistipularis Drive Microbial Community Factors

The relationship between the soil microbial community and soil physical and chemical properties is mutual and complex. Soil microbes play important roles in the depolymerization and mineralization of soil phosphorus [67]. Previous research has found that the soil rhizosphere fungi significantly enhanced photosynthetic performance and the exchange of water and gas [68]. It was found that after inoculation of *Hebeloma crustuliniforme*, the root hydraulic conductivity of *Populus tremuloides* was greater than that of the control seedlings, and the water conductivity increased [69]. We found the same situation in male and female *S. linearistipularis*; this may be related to the *Hebeloma* in male and female plants. Soil rhizosphere fungi can promote the absorption of soluble phosphorus and can also improve the utilization of insoluble phosphorus and organophosphorus by plants. There is a significant positive correlation between the *Tomentella*, *Geopora*, and content of TP in fungal community. *Tomentella* promotes plant nutrient cycling by breaking down understory litter [70]. *Geopora* is defined as an important partner of host plant resistance to drought stress and is abundant in arid and alkaline soil environments [71]. It was also found that *Salix* plants can be highly adapted to salt stress, especially insoluble *Tomentella*, *Geopora,* and *Hebeloma* [72]. These explain that soil fungi play an important role in promoting saline–alkali tolerance among male and female *S. linearistipularis*.

Redundancy analysis (RDA) showed significant differences between soil characteristics and bacterial communities (Figure 6). The effects of S-UE on soil bacterial community composition were significant (*p* < 0.05). The effects of S-UE, SWC, pH, and TN on soil fungal community composition were significant (*p* < 0.05). The above analyses results indicate that pH, TN, SWC, and S-UE were the main environmental factors of microbial community structure in active soil. A strong correlation between soil pH and bacterial community structure of male and female *S. linearistipularis* was observed. The pH can affect the bacterial community structure of *S. linearistipularis* by changing soil environmental factors. Soil urease is an important enzyme that affects the decomposition of organic nitrogen and it is involved in the transformation of nitrogen-containing organic compounds in soil, which are absorbed and utilized by plants [73]. Therefore, the activity of soil urease can be used as an index to evaluate the supply level of soil nitrogen. With the growth and development of *S. linearistipularis*, soil nitrogen is consumed, and soil microbial nitrogen gradually decreases. High temperature and humidity assist the remaining organic materials in the soil in further fermentation and decomposition; then, microbial activity is enhanced, the microbial number increases again, and the decomposition of organic matter increases the nitrogen content in the soil. These results indicate that soil urease could interact with the microbial community structure through the transformation pathway of nitrogen-containing organic matter in the soil. It is known that soil fungi can help plants to access water and nutrients thanks to their mycelium network and uptake capacities, due to fungal natural tolerance towards water fluctuations which, in the end, benefit the plant [74]. This explains that S-UE is the bacterial community, and S-UE, PH, SWC, and TN are the main soil-influencing factors of the fungal community.

## 5. Conclusions

The male and female *S. linearistipularis* in saline–alkali land of Songnen Plain demonstrated sex-specific preferences for soil microbial communities and microenvironments. We found that there were differences in the microbial communities and rhizosphere soil physical and chemical properties of male and female *S. linearistipularis*, and soil physical and chemical properties were driving factors for microbial communities. It was concluded that female plants had a more stable and diversified community structure than male plants. Female plants had strong ability to improve saline–alkali soil for growth. This also provides an experimental baseline to use female willow plants in improved approaches to repairing saline–alkali soils in the future. This study enriches the available data on the physical and chemical properties of *S. linearistipularis* in relation to rhizosphere soil, as well as enriching the data on the plant index determination of willow rhizosphere soil.

## Figures and Tables

**Figure 1 microorganisms-11-02455-f001:**
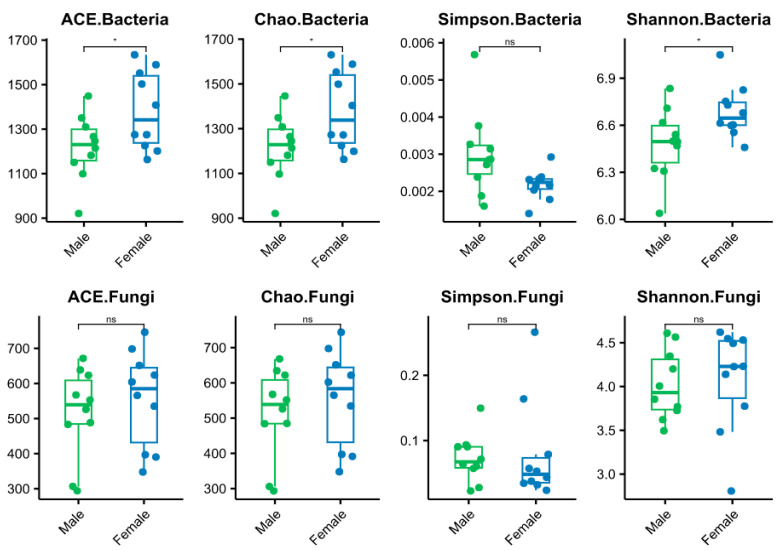
Box plot of soil bacterial and fungal community diversity index of male and female *S. linearistipularis*. There were significant differences between male and female soils, as indicated by (* *p* < 0.05), while ns indicated no significant differences.

**Figure 2 microorganisms-11-02455-f002:**
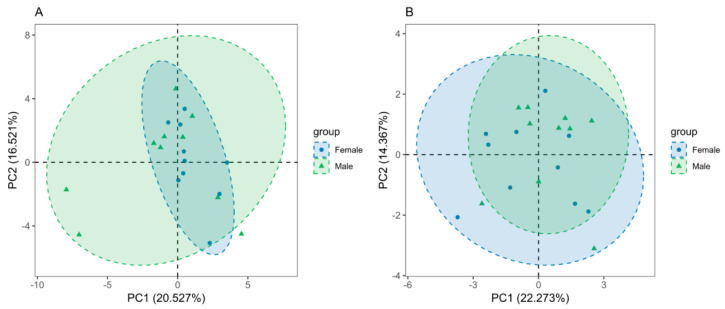
Principal component analysis (PCA) showed that there were differences in the rhizosphere soil and bacterial communities (**A**) and fungal communities (**B**) between male and female *S. linearistipularis*.

**Figure 3 microorganisms-11-02455-f003:**
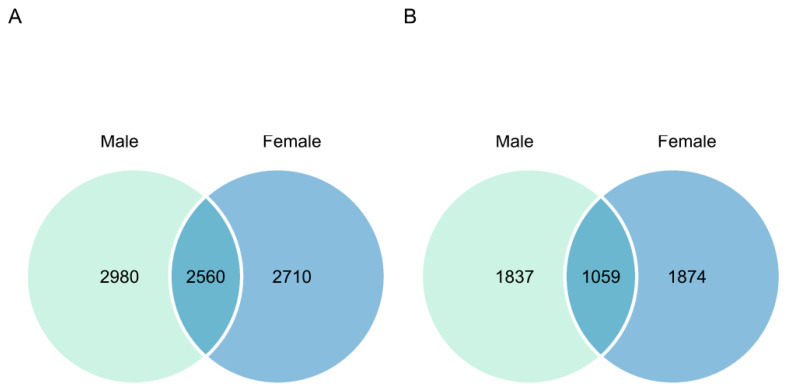
The Venn diagram shows the composition and structural differences of bacterial (**A**) and fungal (**B**) ASV in male and female strains of *S. linearistipularis*.

**Figure 4 microorganisms-11-02455-f004:**
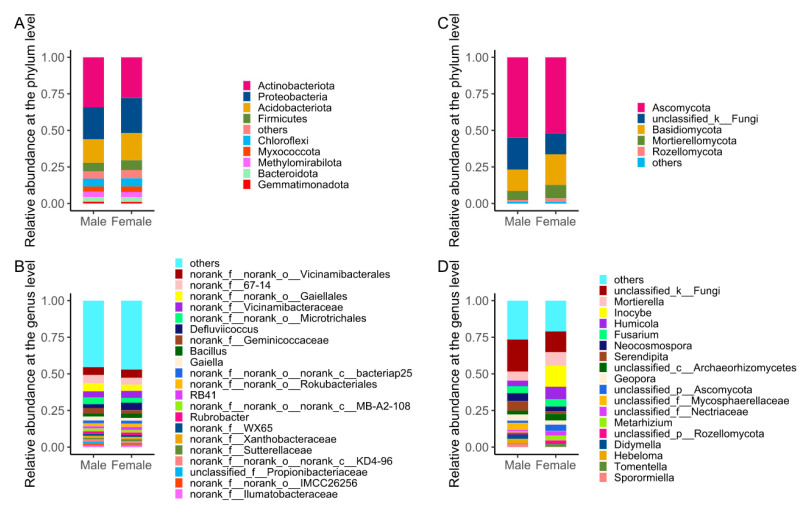
Bacterial abundance of male and female *S. linearistipularis* at phylum (**A**) and genus (**B**) levels; the abundance of fungi in male and female *S. linearistipularis* rhizosphere at the phylum (**C**) and genus (**D**) levels.

**Figure 5 microorganisms-11-02455-f005:**
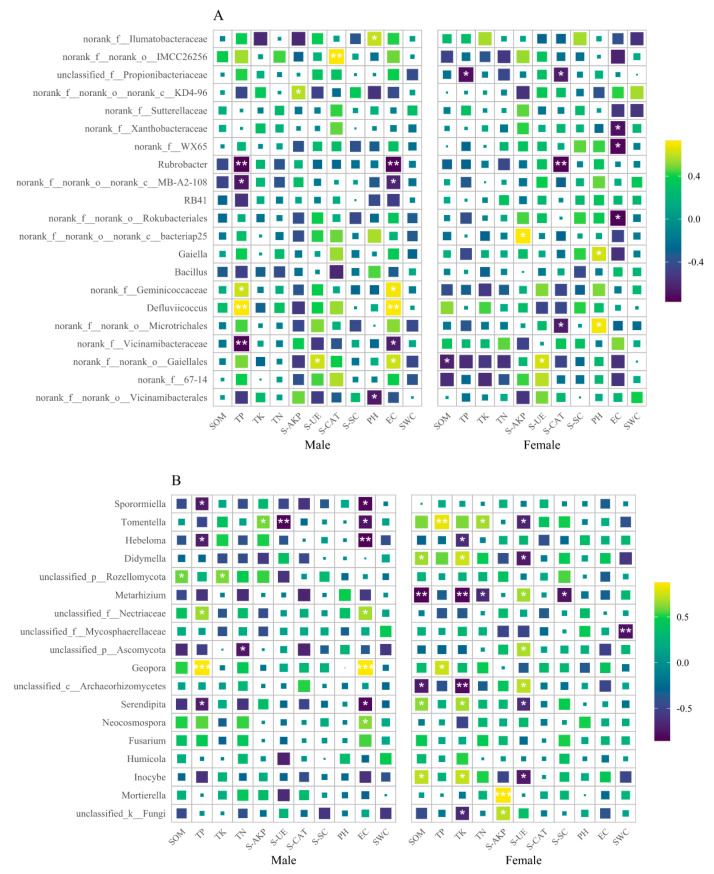
Correlation between soil nutrients and rhizosphere microorganisms genera with relative abundance over 1% ((**A**) indicate genera of bacteria; (**B**) indicate genera of fungi; *—significantly correlated at 0.05 levels; **—significantly correlated at 0.01 levels.

**Figure 6 microorganisms-11-02455-f006:**
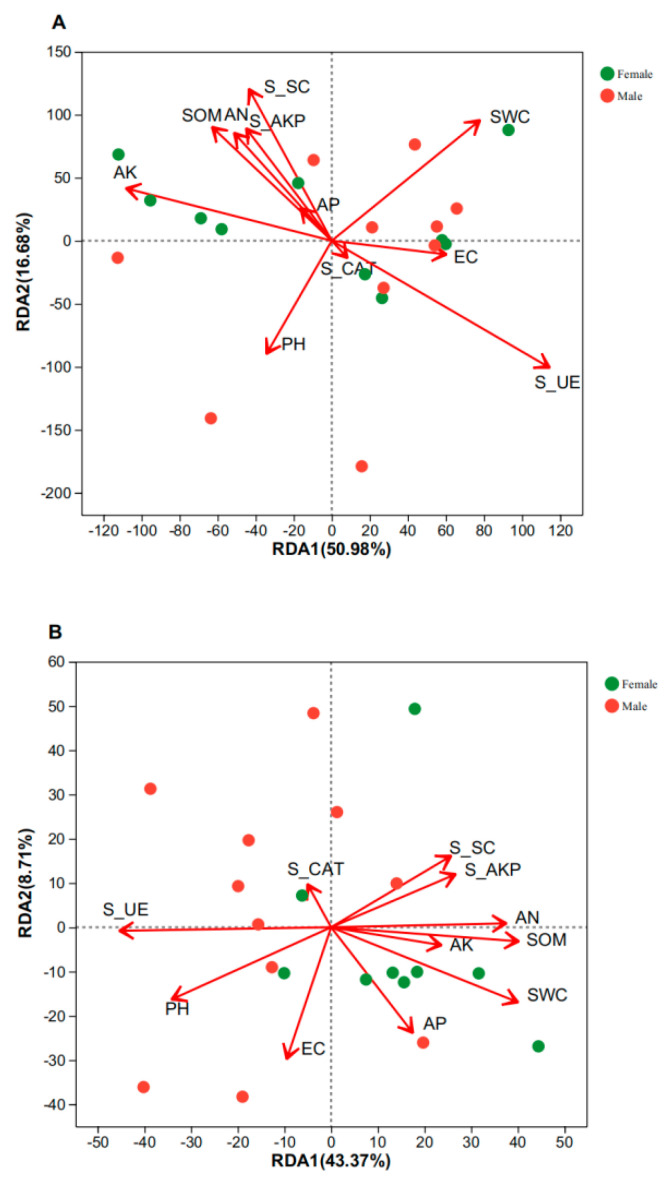
Redundancy analysis (RDA) analysis of bacterial (**A**) and fungal (**B**) community structure and soil physicochemical properties in the rhizosphere soil of *S. linearistipularis*.

**Table 1 microorganisms-11-02455-t001:** The physicochemical characteristics of soil (means± SD).

Soil Properties	Male	Female
SOM (%)	4.52 ± 1.23 b	7.66 ± 1.39 a
TP (mg/kg)	14.12 ± 4.35 a	17.36 ± 7.33 a
TK (mg/kg)	223.52 ± 53.25 a	316.81 ± 202.49 a
TN (mg/kg)	76.93 ± 23.84 b	113.47 ± 19.71 a
S-AKP (μmol/d/g)	19.13 ± 4.29 a	21.25 ± 0.78 a
S-UE (μg/d/g)	712.48 ± 127.40 b	859.23 ± 136.09 a
S-CAT (μmol/d/g)	63.20 ± 0.40 a	63.27 ± 0.27 a
S-SC (mg/d/g)	176.22 ± 47.30 a	203.18 ± 24.39 a
PH	8.43 ± 0.16 a	8.23 ± 0.17 b
EC (us/cm)	191.08 ± 42.71 a	190.73 ± 18.22 a
SWC (%)	22.68 ± 4.23 b	26.90 ± 4.65 a

Different letters after inline values indicated significant differences between male and female plants (*p* < 0.05). Table data are means ± SD (*n* = 10); means refer to the average of index data. Abbreviations: soil organic matter content (SOM), available phosphorus content (TP), available potassium content (TK), available nitrogen content (TN), alkaline phosphatase (S-AKP), urease (S-UE), catalase (S-CAT), sucrase (S-SC), PH, electrical conductivity (EC), and soil water content (SWC).

## Data Availability

Not applicable.

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
