# Peer review of "Effects of Male and Female Strains of Salix linearistipularis on Physicochemical Properties and Microbial Community Structure in Saline–Alkali Soil"

_microorganisms, 2023, doi:10.3390/microorganisms11102455_

Round 1

Reviewer 1 Report

The work presents valuable information on the interaction of the microbiota and the genus of S. linearistipularir plants. The previous studies are morphological and physiological, therefore it is an important approach in the establishment of these plants. The statistical analysis seems very complete to me. The relationships between variables are addressed and the importance of the community structure of the rhizosphere in relation to the genus of the plants is identified. In particular, the high presence of fungi in the substrate and its possible effect on the photosynthetic rate and the efficiency of water use by the host plant.

If this aspect of fungi is expanded, I believe that the article would be further strengthened by finding a higher phosphorus content in female plants and their tolerance with salt.

Author Response

Dear reviewer

Thank you!

Reviewer 2 Report

What are the factors that influence the physical-chemical characteristics of the soil in female plants?

How could you explain the difference in microbial diversity?

There are reports of the components of the root exudates of the male and female plants of S. linearistipularis.

Any control where male and female plants are not exposed to alkaline soils or reports that they exist?

Minor editing of English language required

Author Response

Dear editor

Thank you

Reviewer 3 Report

This research is about sexual differences of Salix linearistipularis in influence the physicochemical properties and rhizosphere microorganisms and its effect on soil salinity. Authors revealed differences between male and female plants and assumed that female plants are more salinity tolerant and are able to improve saline-alkali soil.

However I have comments, because some parts are not clear for me.

Lines 61-64 – this sentence is not clear. It is too large and it is rather complicated to understand the sense. It would be good to divide this one sentence in two or three.

Lines 169-183 should be deleted.

Lines 186-187 are not in agreement with Table 1. The means of TP, TK, S-AKP and S-SC are higher for female plants but statistically there is no difference between the means, what is shown in the table 1. S-CAT are equal for male and female plants.

Line 188 – according Table 1 there is no difference in EC between male and female plants and S-UE is higher in female plants.

Line 210 – the method of evaluating Chao, ACE, Simpson and Shannon indices should be mention in the Material and methods.

Lines 214-216 – Are these means medians or average of indices? It should be mentioned.

Lines 318-320 –give the references for determination of leaf physiology indices of S. linearistipularis. Also give the references of studies on determination o rhizosphere soil physicochemical properties. If there are few studies, it means they exist and references should be given.

Line 320 -324 – If this is the conclusion of this work it should be at the end of the discussion or give the references.

Check the title. It should be “strains of Salix linearistipularis on physicochemical properties.

Check the style of references. References 6 and 20 are the same.

Check the spaces between words thought the text.

Line 11 – ‘we’ should be small.

Line 104 – what is 20?

Lines 117, 458 – it should be pH

Lines 381-383 – I think it is better to use S. linearistipularis

Figure 6 – add A and B in the caption

Lines 297 -300 – it seems that these two first sentences should be fused in one. In this form the sense of the first sentence is lost.

Line 310 – what is this plant? Add the species name. The reader might think of Salix but the speech is about Populus.

line 410 – “Filamnia can symbiosis” should be “Filamina can establish symbiosis”

Lines 472-473 – it should be “Songnen Plain demonstrated sex-specific”.

Author Response

Dear reviewer

Thank you

Round 2

Reviewer 3 Report

The text has been improved.

However, some moments have to be clarified. According to Table 1, only SOM, TN, S-UE, pH and SWC have differences between male and female plants. EC is equal for both groups. It means line 371 is wrong, because there are no differences in S-AKP, S-CAT and S-SC between male and female plants. Also check lines 459 and 460, EC is not lower in female plants, and TP and TK were not higher. The same situation is in abstract. EC is not reduced in female plants.

Check the captions for figures 2, 5, 6.  “Figure 6. Driving factors of soil physicochemical properties of male (Figure.A) and female (Figure.B)” it is unnecessary to repeat ‘Figure’, leave only A and B.

Line 169 - S. linearistipularis needs italic

Line 110 – “The rhizosphere soil was collected in October 2021. Twenty S. linearistipularis were”. Does it mean twenty plants?

Line 277 – “greater than 1%, there were eight in male soils and no seven in female”. On figure 5A I see 7 male and 8 female. And on figure 5 B, I see 9 male and 13 female soils that are correlated with soil nutrients.

Author Response

Dear reviewer

Thank you
